# A mutant with bilateral whisker to barrel inputs unveils somatosensory mapping rules in the cerebral cortex

Nicolas Renier[1]*[†], Chloé Dominici[1], Reha S Erzurumlu[2], Claudius F Kratochwil[3‡], Filippo M Rijli[3,4], Patricia Gaspar[5], Alain Chédotal[1]*

[1]Sorbonne Universités, UPMC Univ Paris 06, INSERM, CNRS, Institut de la Vision, Paris, France; [2]Department of Anatomy and Neurobiology, University of Maryland School of Medicine, Baltimore, United States; [3]Friedrich Miescher Institute for Biomedical Research, Basel, Switzerland; [4]University of Basel, Basel, Switzerland; [5]INSERM, Institut du Fer à Moulin, Paris, France

**Abstract** In mammals, tactile information is mapped topographically onto the contralateral side of the brain in the primary somatosensory cortex (S1). In this study, we describe Robo3 mouse mutants in which a sizeable fraction of the trigemino-thalamic inputs project ipsilaterally rather than contralaterally. The resulting mixture of crossed and uncrossed sensory inputs creates bilateral whisker maps in the thalamus and cortex. Surprisingly, these maps are segregated resulting in duplication of whisker representations and doubling of the number of barrels without changes in the size of S1. Sensory deprivation shows competitive interactions between the ipsi/contralateral whisker maps. This study reveals that the somatosensory system can form a somatotopic map to integrate bilateral sensory inputs, but organizes the maps in a different way from that in the visual or auditory systems. Therefore, while molecular pre-patterning constrains their orientation and position, preservation of the continuity of inputs defines the layout of the somatosensory maps.

*For correspondence: nicolas.renier@icm-institute.org (NR); alain.chedotal@inserm.fr (AC)

Present address: [†] ICM Institute for Brain and Spinal Cord, Hôpital Pitié Salpêtrière, Paris, France; [‡]Chair in Zoology and Evolutionary Biology, Department of Biology, University of Konstanz, Konstanz, Germany

Competing interests: The authors declare that no competing interests exist.

## Introduction

Sensory maps in the brain integrate physical (topographic) and functional constraints. According to the type of sensory modality, these constraints are differently accommodated. In the somatosensory system, the sensory receptors of the periphery establish topographic replicas in the different brain relay stations, in the brainstem, thalamus and cortex, with a size that is roughly proportional to functional importance of the sensory element represented (*Penfield and Boldrey, 1937*; *Woolsey and Van der Loos, 1970*).

Some of the main construction principles of these maps have been elucidated implying collaboration of morphogenetic gradients and neural activity (*Fukuchi-Shimogori and Grove, 2003*; *Pfeiffenberger et al., 2005*; *Rash and Grove, 2006*), but there is continued controversy regarding the degree to which the initial clustering and topographical arrangement of axons carrying the inputs plays a role in map layout (*Erzurumlu and Gaspar, 2012*).

The contributions of molecular pre-patterning and nearest-neighbor clustering can be tested in bilateral sensory maps that integrate bilateral signals where the inputs come from each side of the body and therefore arrive via separate routes. Manipulating the laterality of inputs also provides a method to identify mechanisms of bilateral integration during map-building. This has been tested previously in the visual system (*Rebsam et al., 2009*). However, the natural overlap of the receptive fields for a portion of retinal ganglion cells between the left and right eyes can constrain the topographic organization of the binocular visual cortex. Thus, experiments in the visual system do not

allow easy disambiguation of the different mechanisms at play during establishment of the map. Evidently, there is no such continuity and overlap between the somatosensory receptive fields of the left and right sides of the body. Therefore, if genetic patterning were the main factor controlling integration of bilateral sensory processing in the cortex, one would expect that changing the laterality of a fraction of somatosensory inputs would create the equivalent of a 'binocular' region in the somatosensory cortex. To the opposite, if the map-building rules maintain nearest-neighbor's interactions, one would expect that such manipulation would result in fully segregated representation of bilateral inputs in the cortex. Here, we examined the effect of partial uncrossing of presynaptic afferents to the somatosensory thalamus in the whisker to barrel pathway of mice.

The sensory afferents from the whisker follicles first synapse in the brainstem trigeminal complex and second order neurons in the principal sensory nucleus of the trigeminal nerve (PrV) carry the whisker-specific inputs to the contralateral ventroposteromedial nucleus (VPM) of the thalamus (*Figure 1A*; reviewed in [*Erzurumlu et al., 2010*]). In the present study we focused on a conditional mouse mutant in which *Robo3*, an axon guidance receptor necessary for the crossing of commissural axons (*Sabatier et al., 2004*) is inactivated in rhombomere 3 (*Renier et al., 2010*), the origin of most whisker-specific PrV neurons (*Oury et al., 2006*). The conditional lack of Robo3 only caused a partial crossing defect of trigemino-thalamic axons. Consequently, conditional Robo3 mutants had bilateral sensory afferents in the brainstem to neocortex portion of the whisker-barrel pathway. We found that this resulted in two functional whisker maps in the thalamic relay, VPM, and the barrel cortex, each receiving inputs from a different side of the animal's face. Most interestingly, these maps were entirely segregated, both being confined to the cortical space normally allocated to facial whisker representation, but retained correct orientation and topographic organization. These results suggest that the mechanisms shaping topographic representation of the somatosensory map respect the nearest-neighbor continuity of the peripheral receptor topography, within the position and orientation constraints set by the molecular pre-patterning gradients in the thalamus and cortex.

## Results

### Genetic perturbation of midline crossing signals and emergence of bilateral somatosensory maps

We analyzed the whisker to barrel projection in a previously characterized mouse line (*Renier et al., 2010*) in which the *Robo3* gene has been specifically knocked out in rhombomeres 3 (r3) and r5, using the Krox20 promoter (*Krox20:Cre;Robo3$^{lox/lox}$*, hereafter referred to as *Robo3$^{R3-5}$-cKO*, with *Krox20:Cre;Robo3$^{lox/+}$* referred to as *Robo3$^{R3-5}$-Het*). During development, *Robo3* is transiently expressed in r3 neurons, with expression stopping shortly after axon crossing, suggesting that it does not play a role in later stages of development such as axon targeting (*Badura et al., 2013*; *Michalski et al., 2013*; *Renier et al., 2010*). As expected, in situ hybridization confirmed that *Robo3* expression is deleted from r3 in 12-day-old (E12) *Robo3$^{R3-5}$-cKO* embryos (*Figure 1B*, n = 3/3).

To visualize the trigemino-thalamic pathway, we crossed *Robo3$^{R3-5}$-cKO* mice with *Tau-lox-Stop-lox-mGFP-IRES-nls-lacZ* mice (*Tau$^{GFP}$*) (*Hippenmeyer et al., 2005*). In E12 controls many GFP+ axons cross the midline at the r3 and r5 level (*Figure 1B–D*). The ßgal nuclear reporter showed a dense distribution of Krox20+ cell bodies in the ventral region of the PrV nucleus, as expected from previous fate-mapping experiments (*Oury et al., 2006*). In the *Robo3$^{R3-5}$-cKO;Tau$^{GFP}$* mice, the distribution of ßgal+ neurons and organization of barrelettes was normal, indicating that Robo3 deletion did not alter development of the PrV nucleus (*Figure 1C* and *Figure 1—figure supplement 1*). By contrast, most of the GFP-labeled axons arising from the PrV failed to cross, although they still projected rostrally towards the forebrain (*Figure 1B–D*; n = 3/3). At E13, coronal sections at the level of r3 showed that the density of GFP+ commissural axons at the ventral midline was strongly reduced in the *Robo3$^{R3-5}$-cKO;Tau$^{GFP}$* mice, but also indicated that a subset of axons still crosses (*Figure 1B*). Double staining for NeuN and the nuclear cre reporter ßgal in adult sections of *Robo3$^{R3-5}$-Het;Tau$^{GFP}$* mice (*Figure 1—figure supplement 1*; n = 5) showed that 88% of NeuN+ neurons in the ventral part of PrV express ßgal, but that a small subset of the NeuN+ PrV neurons (10.7 ± 1.5%) were not ßgal+ and probably did not express Cre recombinase.

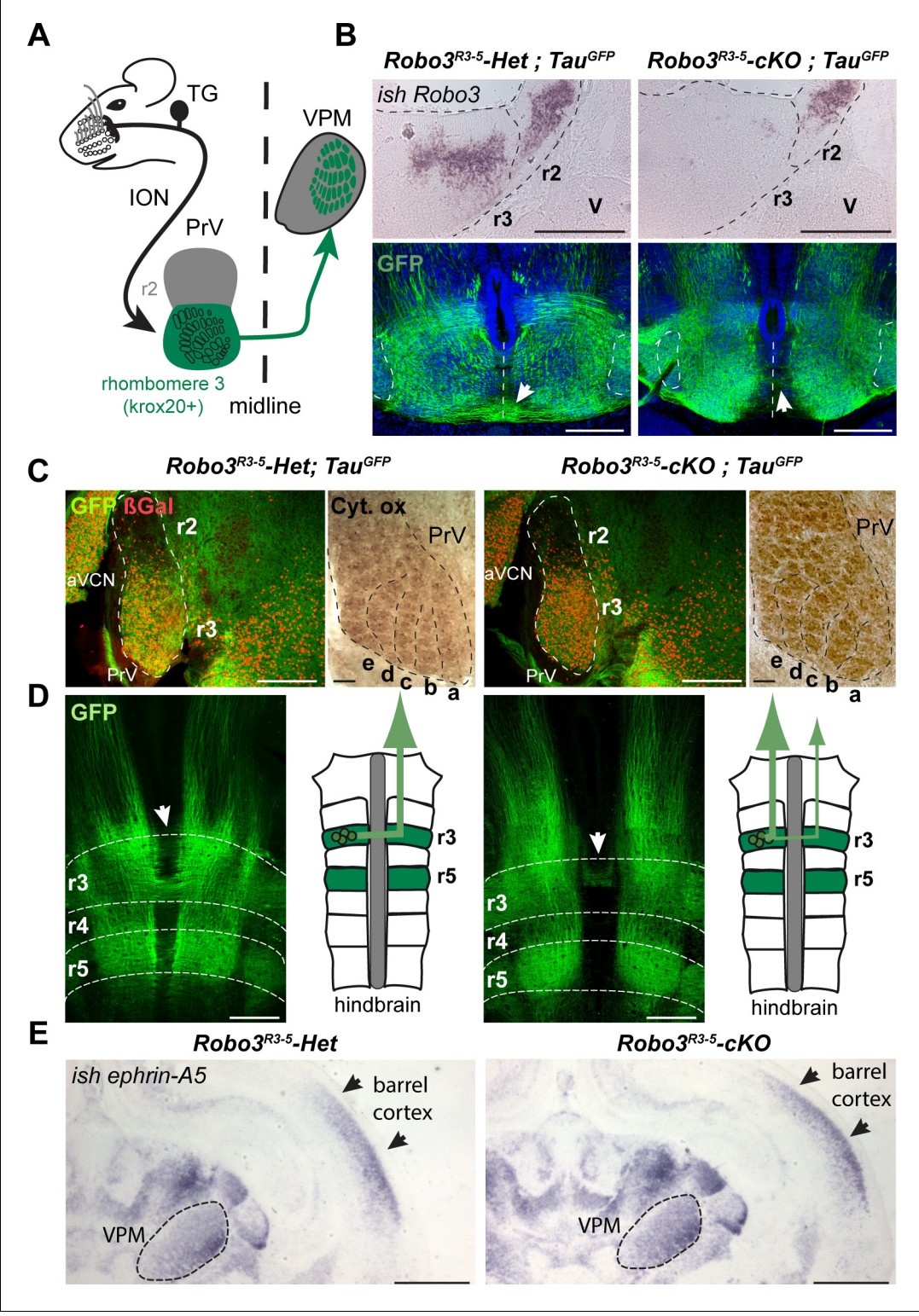

**Figure 1.** Rewiring of r3 and r5 derived hindbrain projections to midbrain/forebrain projections in *Robo3^{R3-5}*-cKO mice. (**A**) Schematic representation of the mouse whisker to barrel somatosensory pathway. (**B**) Top panels: in situ hybridization (*ish*) with a *robo3* probe on coronal sections at rhombomeres 3 and 2 (r3, r2) level in E12 embryos. No staining is observed in *Robo3^{R3-5}*-cKO mice in r3. Trigeminal ganglion (V) neurons do not express *Robo3*. Bottom panels: coronal sections at r3 level in E13 *Robo3^{R3-5}*-Het;*Tau^{GFP}* or *Robo3^{R3-5}*-cKO;*Tau^{GFP}* embryos stained for GFP. GFP+ commissures are strongly reduced in mutants, but a few axons still cross (arrows). (**C**) Cytochrome

*Figure 1 continued on next page*

*Figure 1 continued*

oxidase staining (Cyt. ox.), and ßGal, GFP co-immunostaining of coronal sections of P4 *Robo3$^{R3-5}$-Het;Tau$^{GFP}$* or *Robo3$^{R3-5}$-cKO;Tau$^{GFP}$* brains at the level of the brainstem principal trigeminal nucleus (PrV), showing the barrellettes. Rows a to e are indicated. The barrelette patterns and ßGal+ cell distribution are similar in control and *Robo3$^{R3-5}$-cKO* mutant mice. aVCN: anterior ventral cochlear nucleus. (D) Flat-mount view and scheme of the hindbrain of E12 *Robo3$^{R3-5}$-Het;Tau$^{GFP}$* or *Robo3$^{R3-5}$-cKO;Tau$^{GFP}$* embryos. Commissures are strongly reduced at r3 and r5 levels in mutants but a subset of axons still cross in r3 (arrowheads). GFP+ axons still project rostrally towards the midbrain. (E) Coronal sections at the level of the forebrain VPM thalamic nucleus and barrel cortex of P0 *controls* or *Robo3$^{R3-5}$-cKO* brains hybridized with an *ephrin-A5* probe, showing the expression gradients of the molecule, which are unaffected by the conditional deletion of the Robo3. Scale bars are 400 µm, except *ish* and Cyt. ox. (100 µm) and (E) (500 µm).

The following figure supplement is available for figure 1:

**Figure supplement 1.** Normal organization of the principal trigeminal nucleus (PrV) in *Robo3$^{R3-5}$-cKO* mice.

---

As ephrins and their receptors have been shown to control targeting and orientation of thalamo-cortical projections for visual or somatosensory axons (*Dufour et al., 2003*; *Pfeiffenberger et al., 2005*), we used in situ hybridization to check the pattern of *ephrin-A5* mRNA expression at P0 in controls and *Robo3$^{R3-5}$-cKO* mutants. The expression gradients of *ephrin-A5* in the cortex and thalamus were not noticeably different in the mutants and the controls (*Figure 1E*, [n = 2)], suggesting that deletion of Robo3 in the brainstem did not affect expression of patterning cues in the thalamus and cortex.

Three-dimensional imaging of the trajectory of the trigemino-thalamic (TT) tract from the brainstem to the thalamus using iDISCO (*Renier et al., 2014*) (*Figure 2A*) revealed that at P4, the GFP+ axons had a similar trajectory in the TT tract in both control and mutant mice (n = 5), although the tract appeared slightly more defasciculated in the mutants. GFP+ axon terminals arborized in the VPM in both control and mutant mice and formed barreloids (*Figure 2A,B*; *Video 1*). In all mutants and controls (n = 14 for each genotype), the r3-derived GFP+ (r3-GFP+) axons projected to the dorsolateral VPM containing the barreloids (*Figure 2B*). Abnormal organization of the whisker barreloids was noted in the mutant VPM. In controls, all the barreloid rows coincided with a dense r3-GFP+ axon territory (*Figure 2B*; n = 4/4), whereas in mutants two distinct zones were observed (n = 4/4): a lateral VPM domain containing a high density of GFP+ axons and a medial VPM domain with only sparse patches of GFP+ axons (*Figure 2B*). These two VPM domains contained barreloids, identified by cytochrome oxidase staining, and were of comparable surface area in coronal sections through the middle of the VPM (0.19 ± 0.006 mm$^2$ for the lateral dense GFP+ domain and 0.18 ± 0.003 mm$^2$ for the medial patches of GFP+ domain, p=0.46). Moreover, they were separated by a cytochrome oxidase-free septum. These observations suggest that in the *Robo3$^{R3-5}$-cKO* mice, the VPM is split into two separate domains, each with different barreloid patterning (although the lateral domain contains the highest density of r3-derived projections, and larger barreloids).

We traced the PrV to VPM projections anterogradely in P4 *Robo3$^{R3-5}$-cKO* mice using carbocyanine dyes. In control mice the PrV-VPM projection was completely crossed, whereas in mutants the VPM received bilateral innervation from the PrV (*Figure 2C*). Moreover, the position, shape and size of the traced projections in the VPM was reminiscent of the two domains described previously, suggesting that in the mutants, the dense GFP+ lateral region might correspond to abnormal ipsilateral projections from the PrV, while the medial patches might originate from the contralateral side.

These observations indicate that the mutant VPM receives segregated ipsilateral and contralateral trigemino-thalamic inputs. Retrograde injections from the VPM labeled cell bodies on both contralateral and ipsilateral trigeminal PrV nuclei in *Robo3$^{R3-5}$-cKO* mutants (3.3 times (n = 3) more cell bodies were labeled ipsilaterally than contralaterally) (*Figure 2D*; n = 4/4). Interestingly, the neurons projecting ipsilaterally and contralaterally were mixed in the ventral PrV in the *Robo3$^{R3-5}$-cKO* mutants, in contrast to the segregation of their projections seen in the anterograde tracings.

Overall these data suggest that a large fraction of the rhombomere 3-derived trigemino-thalamic axons project ipsilaterally in *Robo3$^{R3-5}$-cKO* mutants, but that some still project contralaterally either because Cre recombination is incomplete or this occurs after crossing.

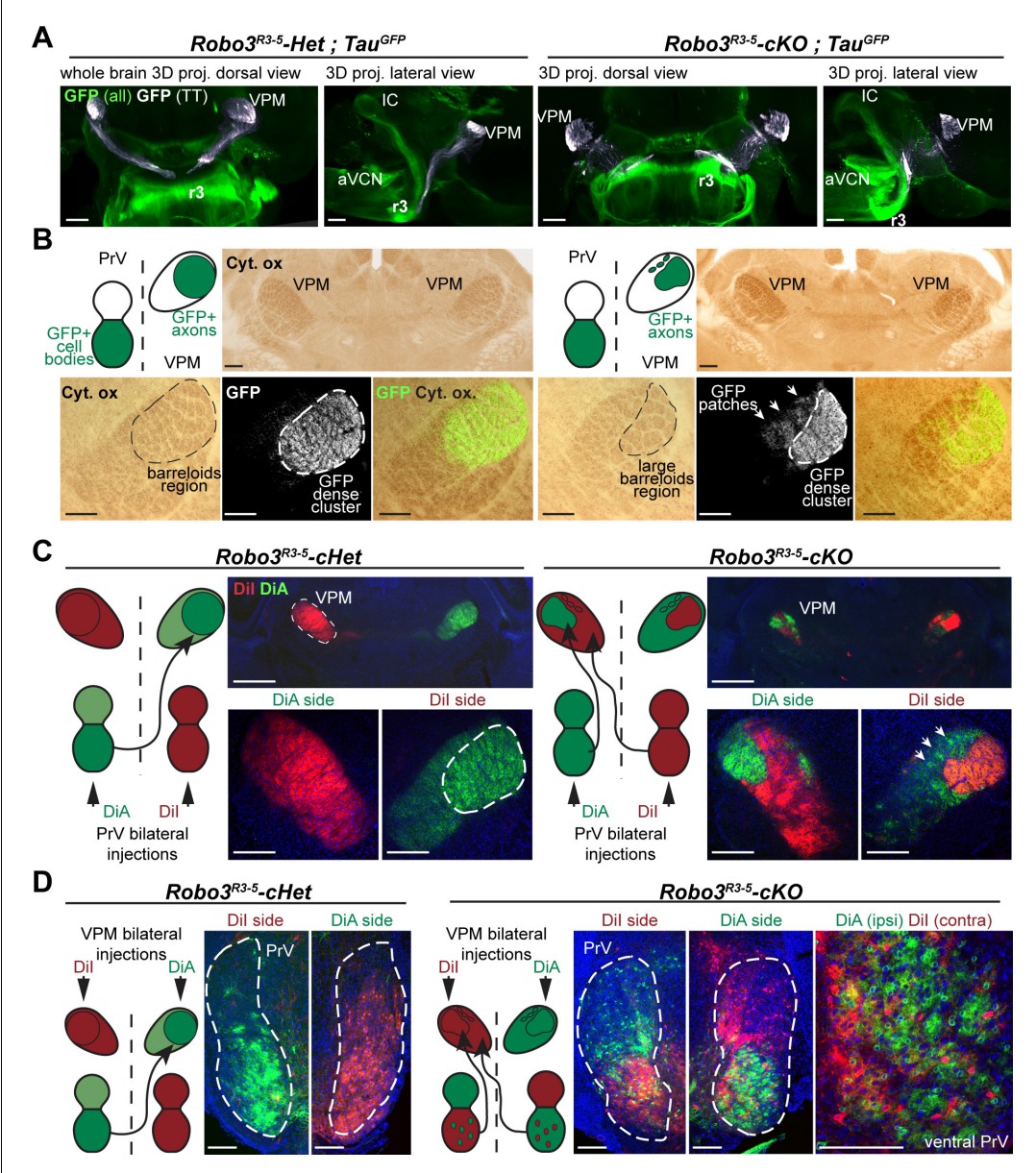

**Figure 2.** Organization of the projections to the VPM. (**A**) Whole-mount immunostaining for GFP in *Robo3^{R3-5}-Het;Tau^{GFP}* and *Robo3^{R3-5}-cKO;Tau^{GFP}* P4 brains cleared with iDISCO. Dorsal projections (left) and lateral projections (right) are shown for each case. The trigemino-thalamic tract is color-coded in gray, with the rest of the GFP signal in green. (**B**) Coronal sections of P4 mouse brain through the sensory thalamus (VPM) stained for cytochrome oxidase. In *Robo3^{R3-5}-Het;Tau^{GFP}* mice, GFP+ axons project to the barreloid area of the VPM. In *Robo3^{R3-5}-cKO;Tau^{GFP}* mice, barreloids are found in two regions, a lateral region containing most of the GFP+ axons and a medial region (arrows) containing only a few patches of GFP axons. (**C**) P4 mice injected bilaterally with DiI and DiA at the level of the PrV nucleus. Sections were collected at the level of the VPM. In controls, the PrV-VPM projection is entirely crossed. In mutants, the VPM receives bilateral inputs from the ipsilateral and contralateral PrV. (**D**) P4 hindbrain cross-sections at the level of the PrV in controls and mutants after unilateral injections of DiA and DiI in the VPM (depicted in the schematics). In controls, PrV trigemino-thalamic projection neurons are labeled by dye injected into the contralateral VPM. In mutants, the dorsal PrV also has only contralaterally labeled neurons, whereas the ventral PrV contains interspersed ipsilateral and contralaterally labeled neurons. Scale bars are 300 µm.

The following figure supplement is available for figure 2:

**Figure supplement 1.** Tract organization and timing of arrival of ipsilateral and contralateral projections from the principal trigeminal nucleus (PrV) to the thalamus in the *Robo3^{R3-5}-cKO* embryos.

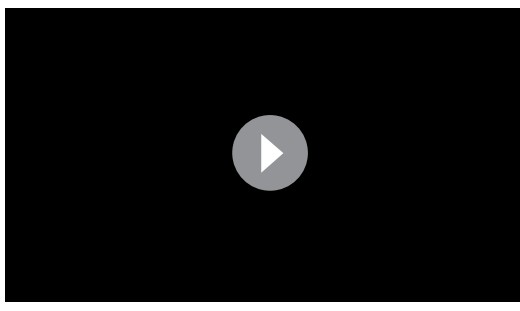

**Video 1.** Rhombomere 3 projections from the brainstem to the VPM. Whole-mount immunostaining for GFP in *Robo3^{R3-5}-Het;Tau^{GFP}* P4 brains cleared with 3DISCO. The GFP is shown in green, and the trigemino-thalamic tract is color-coded in gray.

It is possible that the time of arrival of the ascending axons in the VPM could be a factor in segregation of the ipsilateral and contralateral domains because of shorter paths in the mutants. Ipsilateral axons in mutants may reach the VPM first and occupy the dorso-lateral quadrant of the VPM normally populated by the larger barreloids in controls. We looked at development of the tri-gemino-thalamic (TT) tract at E15.5 in *Robo3^{R3-5}-cKO;Tau^{GFP}* mutant embryos, when axons from the PrV have not yet reached their targets in the VPM (*Figure 2—figure supplement 1A*, n = 3) (*Kivrak and Erzurumlu, 2013*). In this line, both ipsilateral and contralateral PrV projections to the VPM are GFP+. To selectively label contralateral PrV axons, we injected DiI unilaterally into the PrV. In the TT tract, red axons (DiI+ only), yellow axons (double positive for GFP and DiI), and green axons (GFP+) were organized in a medial to lateral gradient (*Figure 2—figure supplement 1A*, n = 3), suggesting that ipsilateral and contralateral axons in the TT tract might be pre-sorted before reaching their target. Ipsilateral (green only) axons were always seen next to contralateral axons (red and yellow) in the TT tract up to the rostral-most sections containing the endings/growth cones of the developing axons. Therefore, ipsilateral axons had no measurable lead over contralateral axons before reaching the VPM at this stage. Indeed, in both *Robo3^{R3-5}-Het;Tau^{GFP}* and *Robo3^{R3-5}-cKO;Tau^{GFP}* mutant E18 embryos, the TT have reached the VPM and the terminals fill the whole dorsal region (*Kivrak and Erzurumlu, 2013*) (*Figure 2—figure supplement 1B*, n = 3). However, the precise timing of arrival of the contralateral and ipsilateral axons in the mutant VPM is difficult to assess. Although we did not find evidence for delayed arrival of the contralateral TT projections in the VPM, we cannot rule out that ipsilateral projections reach their targets earlier and hence have a competitive advantage to innervate the VPM.

To determine the 3D organization of the VPM map in the *Robo3^{R3-5}-cKO* mutants, and the origin of the GFP+ patches, we performed whole-mount imaging of brains from *Robo3^{R3-5}-cKO;Tau^{GFP}* mutants and heterozygous controls at P8 using iDISCO (*Figure 3*)(*Belle et al., 2014*; *Renier et al., 2014*). A suitable angle was determined for optimal projection of the barreloids in the thalamic whis-ker map onto a plane: 3D datasets were oriented at a 45° oblique angle from both coronal and hori-zontal planes (*Figure 3A*, *Figure 3—figure supplement 1A* and *Video 2*). In the *Robo3^{R3-5}-cKO;Tau^{GFP}*mutants, the GFP dense and patched regions were manually segmented to show their respec-tive 3D domains. In both regions, barreloids were organized in rows reminiscent of the control VPM map (*Figure 3A* and *Figure 3—figure supplement 1B*, n = 3).

To determine whether each region corresponded to a distinct whisker map, we performed unilat-eral lesions of the infraorbital nerve (ION) at P0 and the GFP+ projections in the VPM were imaged at P8. In control mice, the unilateral ION lesions caused fusion of the barreloids in the contralateral VPM (*Figure 3B* and *Figure 3—figure supplement 1C*; n = 3). In mutants, barreloid fusion was found in both the ipsilateral and the contralateral VPM: ipsilaterally, in the GFP-dense region, and contralaterally in the GFP-patched region (*Figure 3B*, n = 5). This indicated that, as suggested by the tracing experiments, the GFP-dense region in the VPM receives ipsilateral inputs from the PrV, whereas the GFP-patched region receives contralateral inputs. This also demonstrated that the GFP-patched region carries somatosensory inputs from the infraorbital branch of the trigeminal nerve. Moreover, in addition to fusion of the barreloids, the thalamic map that sustained sensory depriva-tion was reduced in size, while the adjacent non-deprived map was enlarged (*Figure 3C*, n = 5). This showed that sensory-activity-based competition defines the final space allocated to each map in the mutant VPM.

To verify whether both maps in the VPM received inputs from the periphery, we performed intact brain c-Fos immunolabeling in control and mutant mice whose whiskers were shaved on the left side, and b, d rows were spared on the right side (n = 3). The intact brain immunolabeling gave us the opportunity to navigate the complex 3D organization of the VPM using arbitrary oblique

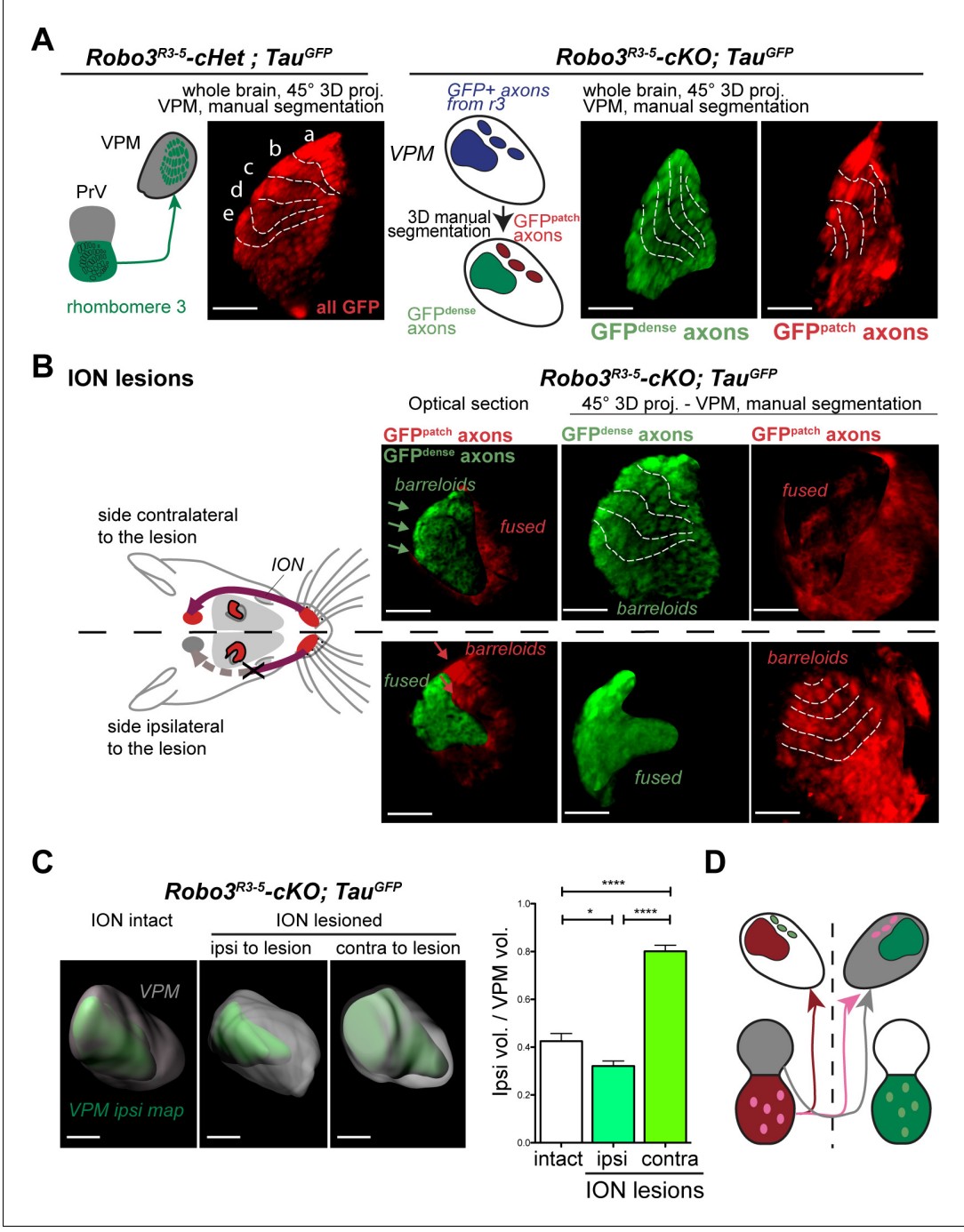

**Figure 3.** Structure of the VPM maps revealed by infraorbital nerve (ION) lesions. Whole-mount scans of 3DISCO cleared P8 *Robo3^R3-5^-Het;Tau^GFP^* and *Robo3^R3-5^-cKO;Tau^GFP^* brains immunostained for GFP. Optical sections and 3D oblique projections are presented. (**A**) Control and mutant brains, ION intact. The oblique projection reveals the topographic barreloid organization in controls (left panels) or mutants (right panels). In the mutant, the dense and patched domains of GFP+ axons (green and red respectively) were manually segmented. (**B**) Mutant brains, unilateral ION lesions. The data are presented as in (**A**). On the side contralateral to the lesion, the barreloids in the patched projection map (red) are fused (n = 3), while the topographic organization of the barreloids in the dense GFP+ domain is still visible (green). The opposite is seen on the side ipsilateral to the lesion: the dense domain of GFP+ axons (green) reveals a fused map while rows of barreloids are visible in the patched domain (red). (**C**) Expansion and retraction of the VPM domains in P8 *Robo3^R3-5^-cKO;Tau^GFP^* brains after lesions. (**D**) Model of the VPM organization in mutant mice deduced from the lesion experiments. Scale bars are 300 μm.

*Figure 3 continued on next page*

*Figure 3 continued*

The following figure supplement is available for figure 3:

**Figure supplement 1.** 3D manual segmentation of the VPM and description of the projection plane.

projection planes (*Figure 4A*). In controls, two bands of c-Fos+ cells were seen contralateral to the spared whiskers, revealing the barreloids in rows b and d (*Figure 4B*). In *Robo3^{R3-5}-cKO* mice, a dual band pattern in the VPM was seen on both sides of the brain. In the ipsilateral VPM, the bands were visible on the same intersecting plane as controls. On the contralateral VPM, the bands of activity were visible on a more medial plane, at the edge of the VPM annotation.

Overall these experiments provide a model for organization of the VPM in *Robo3^{R3-5}-cKO* mutant mice (*Figure 3D* and *Figure 4C*). In the mutant VPM, sensory inputs from the ipsilateral PrV establish a dorsolateral map, with dense projections. Adjacent to this map, inputs from the contralateral PrV project to the dorso-medial VPM as discrete patches. These two thalamic maps are organized in rows, reminiscent of the normal thalamic map. Our finding that whisker stimulation triggers activity-related expression in the two VPM maps in the mutant suggests that the barreloid organization is functional (*Figure 4C*).

## Formation of bifacial cortical maps

Next, we determined how organization of the VPM in the mutant influences formation of the somatosensory map in the cerebral cortex. Tangential sections through layer 4were stained for cyto-chrome oxidase and Vglut2 immunoreactivity to label thalamocortical afferents (*Nahmani and Erisir, 2005*) (*Figure 5A* and *Figure 5—figure supplement 1A*). A striking abnormality in the layout of tha-lamic afferents was noted in the posteromedial barrel subfield (PMBSF, which corresponds to the representation of the large whiskers) of S1 in *Robo3^{R3-5}-cKO* mice (n = 5/5; *Figure 5A*). Large barrels were extra-numerous (52 ± 2 barrels in mutant PMBSF *vs.* 33 ± 0 in controls), and were reduced in size (0.04 mm$^2$ ±0.01 per barrel in mutants, compared with 0.09 mm$^2$ ±0.02 in controls, p<0.0001). Moreover, they were arranged into eight rather than the usual five whisker rows, with clear delinea-tion of two separate cortical zones, a central zone and a peripheral zone, each containing distinct barrel rows (*Figure 5A* and *Figure 5—figure supplements 1–3*). These abnormalities were similar in both hemispheres and at all ages analyzed (with only slight individual variations; n = 25/25; *Fig-ure 5—figure supplements 1–3*). To map the functional whisker representation in this unusual map, we monitored activation of the immediate early gene c-Fos following a 1-hour exposure to an enriched sensory environment (*Staiger et al., 2000*). In mice with unilateral trimming of the whiskers (*Figure 5B*), strong c-Fos labeling is normally observed only in the S1 contralateral to the intact whiskers. In *Robo3^{R3-5}-cKO* mice with unilateral whisker trimming, c-Fos was activated in the PMBSF of both hemispheres (*Figure 5B*). Contralateral to the intact whiskers, c-Fos activation was visi-ble in the peripheral barrel rows (contra domain; *Figure 5B*). Ipsilateral to the intact whiskers, a mirror image was noted with c-Fos activation in the central barrel rows (Ipsi domain; *Figure 5B*). In both contralateral and ipsilateral patterns, c-Fos+ cells were detected in all layers from the columns of the stimulated barrels (*Figure 5B-Figure 5—figure supplement 3B*). These results show that in *Robo3^{R3-5}-cKO* mice, the crossed and uncrossed trigemino-thalamic inputs are mapped as two segregated domains, with the ipsilateral map nested within the contralateral map (*Figure 5B*).

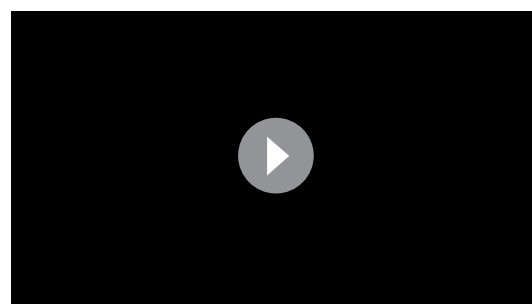

**Video 2.** Segmentation of the VPM in *Robo3^{R3-5}-Het; Tau^{GFP}* mutants. Whole-mount immunostaining for GFP in *Robo3^{R3-5}-Het;Tau^{GFP}* P8 brains cleared with 3DISCO. The GFP dense cluster is segmented in green, and the GFP+ patches are color-coded in red.

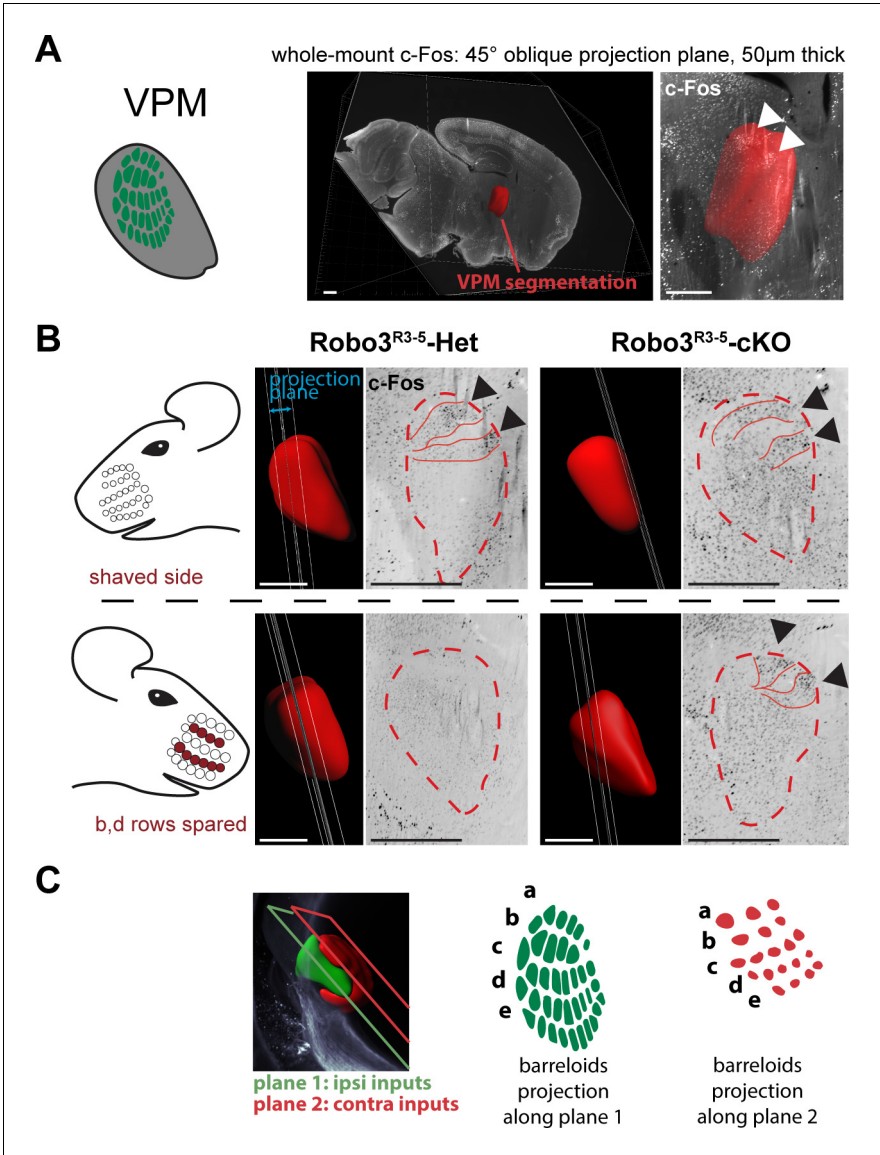

**Figure 4.** Bilateral inputs to the VPM in *Robo3$^{R3-5}$-cKO* mice. Whole brain iDISCO+ scans from adult mice immunolabeled for c-Fos. The whiskers were shaved on the left side, and b, d rows were spared on the right side. (A) Presentation of the projection plane used in the following panels: a 45° oblique (from both coronal and sagittal) 50 μm projection plane was positioned to intersect with the VPM annotation (in red). (B) Details of the c-Fos pattern in the VPM of *Robo3$^{R3-5}$-Het* and *Robo3$^{R3-5}$-cKO* mice on each side. In controls, two bands of c-Fos+ cells were seen on the side contralateral to the spared whiskers, revealing the b and d row barreloids. In *Robo3$^{R3-5}$-cKO* mice, a dual band pattern in the VPM was seen on both sides of the brain. On the ipsilateral side, the bands were visible on the same intersecting plane as in the controls. On the contralateral side, the bands of activity were visible on a more medial plane, at the edge of the VPM annotation. (C) Representation of the VPM organization in the *Robo3$^{R3-5}$-cKO* mice. Scale bars are 500 μm.

## Orientation and polarity of the maps

The segregation of the two maps led to the question of their topographic organization. Two scenarios are possible: (i) the thalamic afferents follow topographic molecular guidance cues expressed in the cortex, with complementary receptor expression in the thalamus; in this case one would expect that neighboring whiskers of the ipsi- and contralateral maps lie in register with one another; (ii) the thalamic afferents are clustered following sensory activity-based rules leading functionally

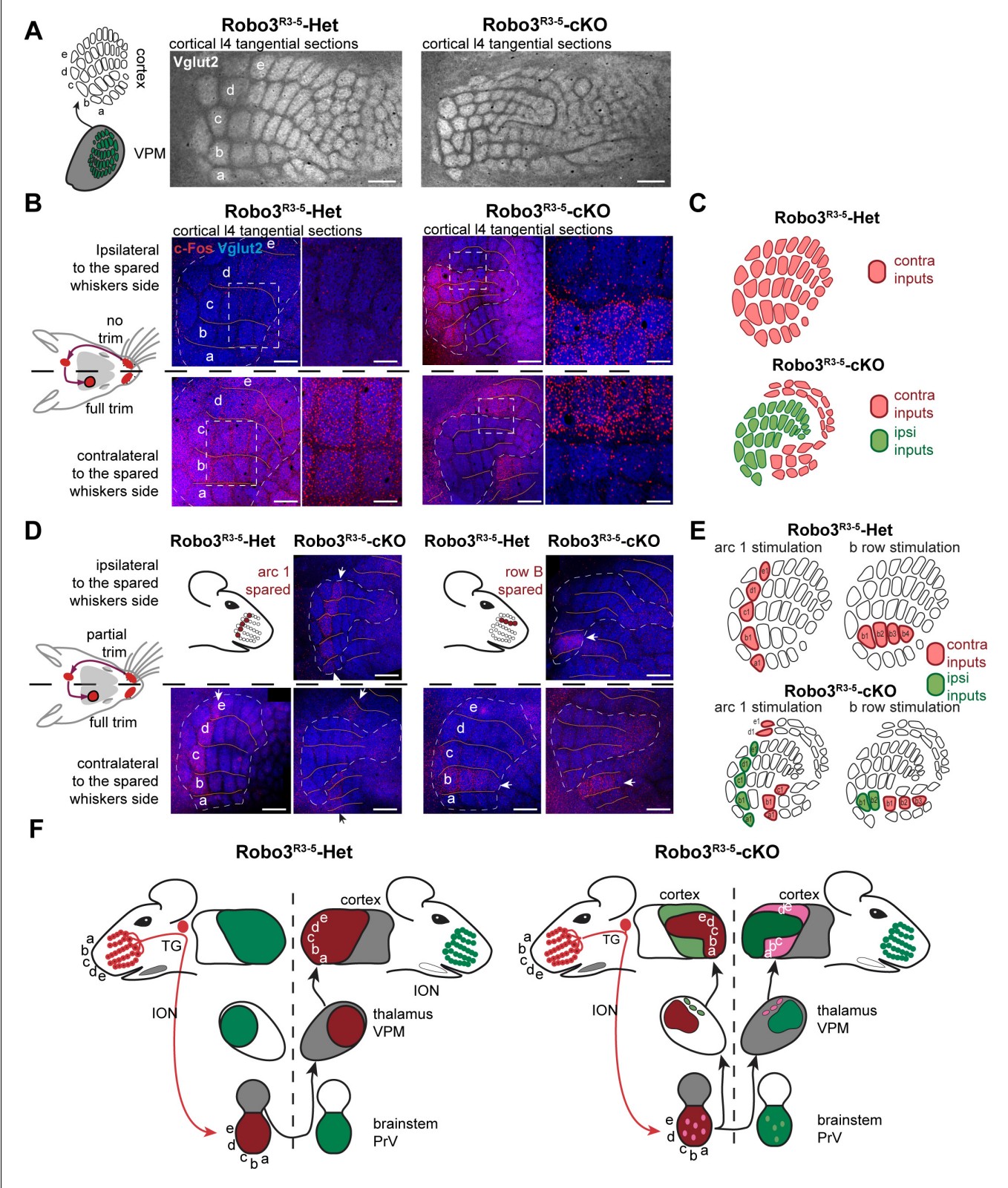

**Figure 5.** Bilateral inputs to the barrel cortex in *Robo3^R3-5-cKO* mice. (**A**) Tangential sections through the barrel cortex from P10 mice stained for anti-Vglut2. Barrels are more numerous and smaller in mutants. (**B**) Tangential sections through Flat-mounted cortices at the level of the barrel cortex in whisker-deprived adult mice immunostained for Vglut2 and c-Fos. In controls, c-Fos+ cell density is high in the barrel cortex contralateral to the intact whiskers and low on the ipsilateral side. In *Robo3^R3-5-cKO* mutants, c-Fos expression is induced bilaterally in complementary domains on either side of

*Figure 5 continued on next page*

*Figure 5 continued*

the cortex, ipsilateral and contralateral to the stimulated side. (C) Interpretation of the results from (B). (D) Tangential sections through Flat-mounted cortices at the level of the barrel cortex, in whisker-deprived adult mice immunostained for Vglut2 and c-Fos. The left side of the face was fully shaved, while only the first arc (left panels) or b row (right panels) was spared on the right side. Only the contralateral sides are shown for controls. Mutants show bilateral patterns of c-Fos (E) Schematic representation of the whisker map deduced from c-Fos activation patterns. (F) General model for the wiring of the *Robo3^{R3-5}-cKO* mutant mice. Scale bars are 200 μm.

The following figure supplements are available for figure 5:

**Figure supplement 1.** Comparison of the organization of the layer 4 of the Barrel Cortex across several adult mutant and controls.

**Figure supplement 2.** Quantification of the surface of the barrels from sections in adult controls and mutants.

**Figure supplement 3.** Quantification of the cross-section surface of individual barrels in cortical layer 4

coordinated afferents to cluster together; in this scenario, the topographical rule of near neighbors would prevail over molecular gradients. To analyze the topographic alignment of the crossed and uncrossed somatosensory maps, we monitored c-Fos expression in S1 after clipping all whiskers except one row or one arc of whiskers on one side (*Figure 5C* and data not shown). When the five posterior-most whiskers of the whisker pad (A1-E1; *Figure 5C*) were left intact in control mice, this resulted in activation of c-Fos in a caudal arc of five barrels exclusively in the contralateral S1 (*Figure 5C*). Likewise, when the second whisker row (B1-B4; *Figure 5C*) was left intact, the corresponding row of barrels was activated in the contralateral S1 (*Figure 5C*). In mutants, c-Fos activation was bilateral, with labeling in both the central (ipsi) and peripheral (contra) PMBSF domains (*Figure 5C*). The general orientation of the two nested maps was similar and resembled that of control mice, likely because of the patterning activity of morphogens that determine the polarity of the map (*Fukuchi-Shimogori and Grove, 2001*, *2003*). However, unlike the visual or auditory bilateral maps, we observed that there were discontinuities in organization of the bilateral somatotopic map, such as a lack of topographic proximity of the ipsi/contra representation for a given barrel or barrel row. Rather, there appeared to be clear separation and independence of the ipsi- and contralateral inputs. Taken together, these results favor the hypothesis that nearest-neighbor's interactions prevail to some extent over the molecular pre-patterning to organize continuous representation of the periphery for each map. However, the molecular gradients still contribute to maintain the general orientation of the maps.

## Competition between ipsi- and contralateral inputs for cortical space

In the mouse models studied here, the space occupied by the barrel map was not increased in mutants ($2.32 \pm 0.05$ mm$^2$ in controls *vs* $1.90 \pm 0.12$ mm$^2$ in mutants, p=0.04), unlike results from other mouse models with duplication of the S1 map where a second S1 map is formed at the expense of other cortical areas (*Fukuchi-Shimogori and Grove, 2001*, *2003*). This suggests that the ipsi- and contralateral thalamic inputs compete to occupy a defined cortical space in S1. Accordingly, individual barrels in mutants were roughly half the size ($44 \pm 2\%$) of those in controls. As with the VPM, we looked again at the consequences of unilateral deprivation of whisker inputs induced by a neonatal (P1) lesion of the ION (*Waite and Cragg, 1982*). In control mice, barrel fusion was observed in the S1 contralateral to the lesion (*Figure 6A*; n = 3/3). In mutants, barrels fused in the maps corresponding to the ipsi- and contralateral representations of the lesioned whisker pad (*Figure 6A*; n = 5/5). Furthermore, the representation of the unlesioned side expanded at the expense of the fused-map. This suggests that there is sensory activity-dependent competition for cortical space between the ipsilateral and contralateral sensory inputs in the mutant S1.

Finally, we checked whether the ipsilateral and contralateral maps were functionally isolated. We took advantage of the ClearMap pipeline (*Renier et al., 2016*) to compare c-Fos activity patterns in the whole brain in an unbiased way, in the unilateral whisker stimulation protocol (n = 3 per group) (*Figure 7A* and *Figure 7—figure supplement 1*). We looked for brain regions that exhibited left-right differences that were opposite in controls and mutants. As expected, the barrel cortex and VPM exhibited statistically significant differences between the shaved and stimulated sides of the

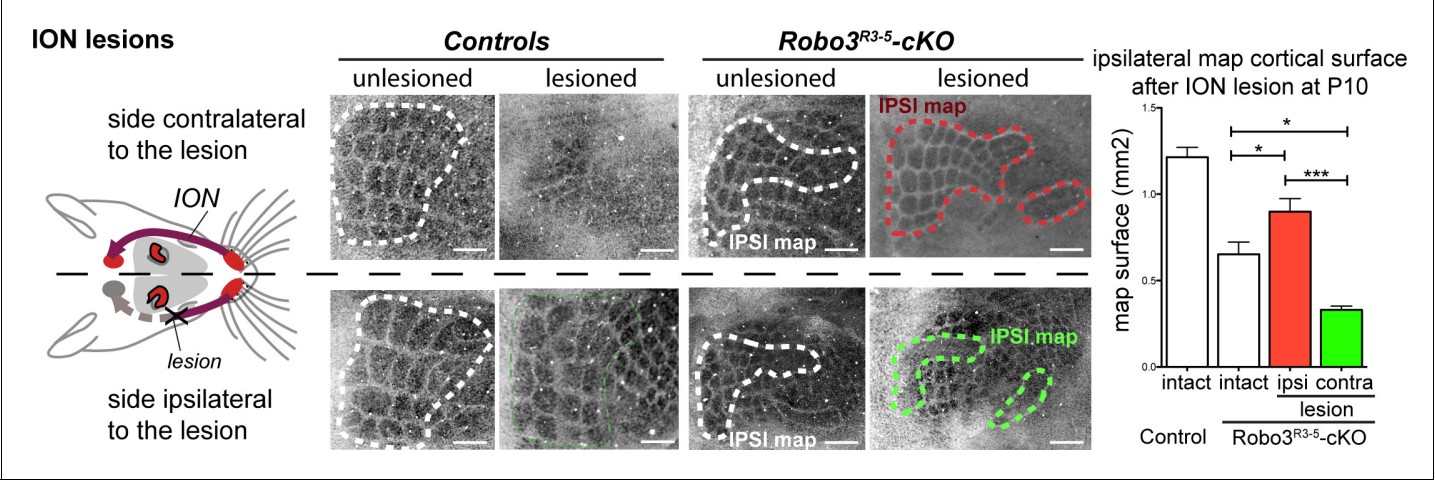

**Figure 6.** Activity-dependent competition between ipsilateral and contralateral inputs in mutant barrel cortex. Tangential sections of controls or *Robo3^R3-5^-cKO* flat-mounted P10 cortices stained for cytochrome oxidase in control conditions or after unilateral lesion of the infraorbital nerve (ION) at P1. In *Robo3^R3-5^-Het* controls, the barrels do not form in S1 contralateral to the lesion, whereas a normal map is seen on the ipsilateral side. In *Robo3^R3-5^-cKO* mutants, contralateral to the lesion, barrels form in the domain processing ipsilateral inputs, and ipsilateral to the lesion , a barreless region is noted in the domain processing ipsilateral inputs. The size of the ipsilateral map is indicated to compare with the maps in unlesioned mutants and controls. Quantification of the surface occupied by the large whisker barrels is shown on the right side. Scale bars are 200 µm.

brain, that were opposite between controls and mutants (*Figure 7A*). Of note, the column of activity detected in d row registered precisely in the same position in control and mutants, showing that the absolute position of the d row in the brain is the same in the aberrant ipsilateral map of the mutant mice a in controls. We then isolated c-Fos+ cells from the upper cortical layers (n = 4). In upper layers, contrary to layer 4, activity patterns are not restricted to the stimulated barrels (*Figure 7B*), but expand over adjacent barrels, due to downstream cortical integration (*Kaliszewska et al., 2012*; *Peron et al., 2015*). We looked at the effect of a patterned sensory deprivation created by trimming rows b and d on one side and shaving all whiskers on the other side. Expansion of c-Fos+ cells was observed within both maps between activated rows in the upper cortical layers 2–3 (*Figure 7B*), but not across the boundaries of each map. This suggests that the ipsilateral and contralateral whisker maps have little to no direct horizontal integration in the upper cortical layers.

## Discussion

In this study, we show that uncrossing a sizeable fraction of the trigemino-thalamic axon tracts results in unexpected anatomical and functional organization in the thalamus and neocortex: a duplication of the facial whisker representation with two segregated maps sharing the same cortical space allotted to somatosensory function. If targeting of the ascending axons was solely organized in a point-to-point manner by patterning gradients in the thalamus and cortex, this genetic manipulation should have resulted in formation of an interspersed 'biwhisker' representation of the whiskers in both thalamic and cortical relays. Instead, we observed complete segregation of the ipsi- and contralateral whisker maps, with each map following the spatial continuity of inputs from the whiskers. However, each map retained correct orientation and topographic organization. This shows that genetic pre-patterning and preservation of the continuity of inputs interact to control, respectively, the position and layout of the somatosensory maps. The absence of left-right mixing of inputs in the *Robo3^R3-5^-cKO* mutant somatosensory cortex also suggests that mechanisms allowing integration of bilateral inputs in the visual cortex might be absent from the somatosensory cortex.

The effect of uncrossing commissures was studied in the same conditional Robo3 mutant in other systems: the olivo-cerebellar projections (*Badura et al., 2013*) and the auditory projections from the cochlear nucleus to the medium nucleus of the trapezoid body (*Michalski et al., 2013*). In these two studies, it was found that affecting the laterality of the projections did not affect the topographic

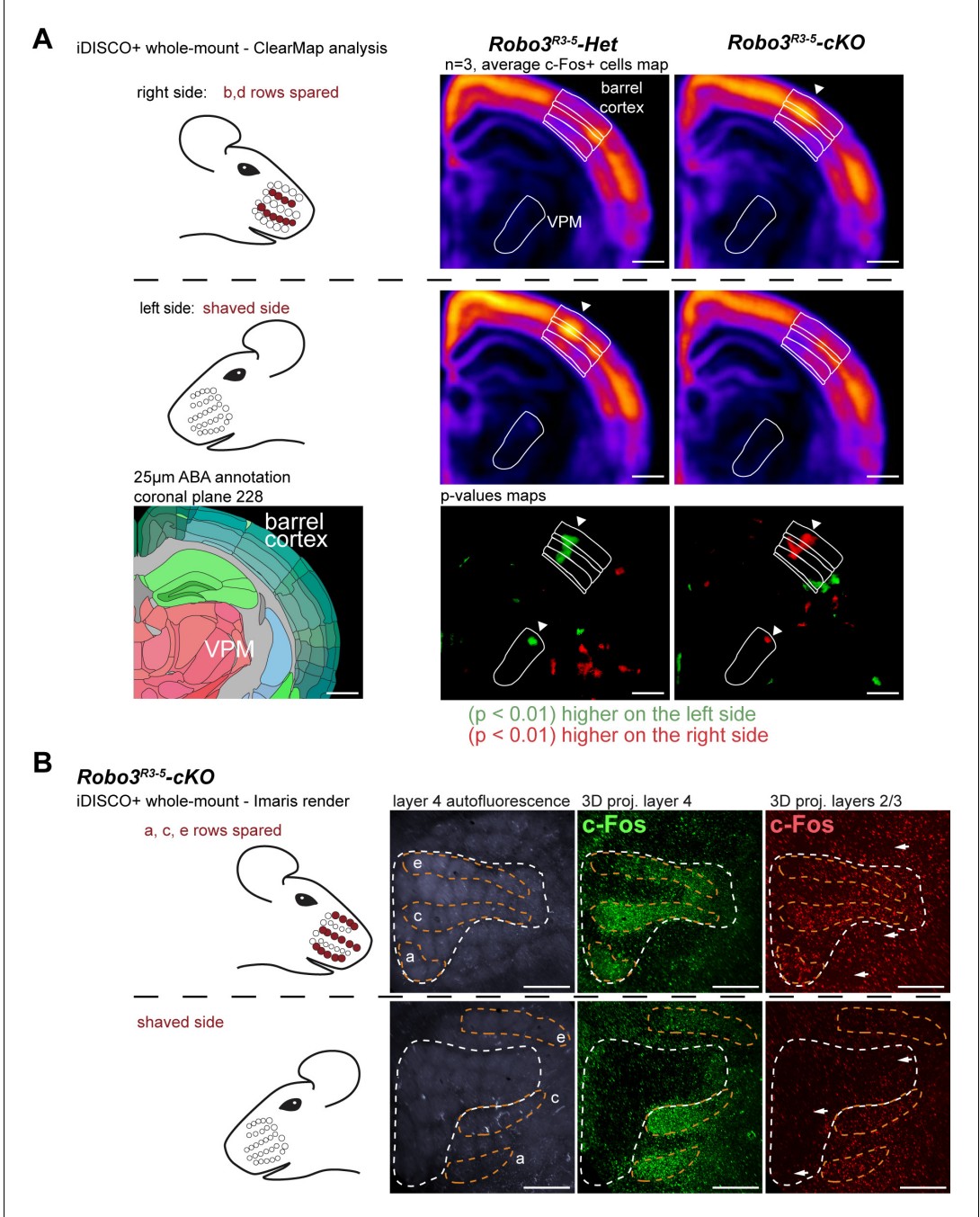

**Figure 7.** Cortical integration of sensory information. (**A**) ClearMap analysis of the c-Fos patterns in iDISCO+ cleared brains in control and mutant mice after 1 hour of exploration of a new environment (n = 3 for each group). The whiskers were shaved on the left side, and b, d rows were spared on the right side. Heatmaps present averaged c-Fos+ cell densities on both sides for three brains for each group, and the p-value maps present the statistically different voxels between the left and right sides, in green when the left side is more active, in red when the right side is more active. At the level of the barrel cortex, as expected, activation was reversed between control and mutant maps in both the VPM and cortex, at the level of the d row (arrowheads). (**B**) iDISCO+ whole-mount c-Fos immunostaining and imaging of adult brains after unilateral stimulation of the rows a, c and e, manually segmented by cortical layers. The pattern in the lower right panel shows spread of c-Fos+ cells between active rows, but no spill-over of activity from the contralateral map to the adjacent ipsilateral map (arrows show blank rows in layers 2/3). Scale bars are 400 μm.

The following figure supplement is available for figure 7:

**Figure supplement 1.** Cortical neuronal activity elicited by an exploration task in adult control and mutants mice.

targeting of the uncrossed axons. However, in the case of the calyx of Held, maturation of the uncrossed synapses was delayed (*Michalski et al., 2013*). It is unclear whether midline crossing changes the molecular expression profile of the axons to promote synaptic maturation, or whether this is an indirect effect of incorrect integration within an otherwise normal network. It would be of value to determine the physiological properties of the ipsilateral map, especially because this new model provides an opportunity to study cortical integration of an abnormal circuit.

The absence of bilateral integration between the two maps in the cortex of *Robo3* mutants differs strikingly from that observed in the binocular region of the visual cortex (*Sato and Stryker, 2008*). The particular organization of the two embedded whisker maps emphasizes an important characteristic of the somatosensory system which combines two different mapping rules: the first being the continuous topographic representation of the body surface, and the second being organization into distinct functional units. Organization of bilaterality in the somatosensory cortex occurs differently when forced in the *Robo3^{R3-5}-cKO* mice than in the normal visual cortex. This raises the tantalizing possibility that specific molecular and activity-based mechanisms absent from the somatosensory system are present in the visual system to promote integration of bilateral information in the cortex, and could therefore be evolutionary mechanisms governing how sensory information is processed in *Bilateria*.

### Orientation of representations and cortical plasticity

Contrary to previous observations of experimental map duplication (*Fukuchi-Shimogori and Grove, 2001*), the present orientation of the two whisker maps was similar in the general rostrocaudal and mediolateral axes, indicating that matching gradients of guidance molecules and their receptors was most likely unchanged, which was confirmed for ephrin-A5 (*Figure 1E*). This contrasts with the mirror image organization of the sensory maps obtained when inducing novel sources of molecular gradients in the somatosensory cortex (*Fukuchi-Shimogori and Grove, 2001*). This also contrasts with observations of map duplication in the visual system, caused by a change in the retinal axon crossing at the midline (*Petros et al., 2008*; *Rebsam et al., 2009*), by lack of one eye (*Trevelyan et al., 2007*) or by lack of one hemisphere (*Muckli et al., 2009*).

The normal orientation of the ipsilateral map in the mutants is surprising. One might have expected that switching laterality of the normally crossed projections would flip the axis of the ipsilateral map from the contralateral map to account for the chiral organization of the left and right sides of the face. As both maps in the mutants respect the normal orientation, the ascending tract from the ipsilateral side may either undergo torsion en route to the VPM to correct the orientation based on molecular gradients present in the lemniscal pathway. Alternatively, the correction of the orientation may occur only at the target site in the VPM based on the gradients of expression of Eph receptors/ephrin ligands in a process akin to the visual tectum (*Feldheim et al., 1998*; *Tessier-Lavigne, 1995*). If true, this hypothesis implicates the presence of additional mechanisms of axonal pruning and refinement, as seen during the post-targeting development of visual projections (*Nakamura and O'Leary, 1989*; *Simon et al., 2012*) to correct the final orientation of the ipsilateral map.

In conclusion, although the initial wiring of the brain relies largely on genetically encoded processes, our results illustrate the remarkable plasticity of the mammalian brain and its ability to accommodate changes in afferent wiring in evolution to create new maps and bilateral representations. These results illustrate also the brain's ability, in the context of developmental brain disorders, to compensate for major axon guidance defects that otherwise would lead to severe brain dysfunction (*Jen et al., 2004*; *Muckli et al., 2009*; *Williams et al., 1994*).

## Materials and methods

### Mice

All animal procedures were carried out in accordance with institutional guidelines (UPMC, Charles Darwin ethic committe and INSERM). Mice were anesthetized with ketamine (Virbac) and xylazine (Rompun). The day of vaginal plug is embryonic day 0 (E0) and the day of birth corresponds to postnatal day 0 (P0).

The Robo3 conditional knockout, *Krox20:Cre* knock-in and *Tau*<sup>GFP</sup> lines were as previously described (*Hippenmeyer et al., 2005*; *Renier et al., 2010*; *Voiculescu et al., 2000*). Unless otherwise mentioned, controls were *Robo3*<sup>lox/lox</sup> or *Krox20:Cre;Robo3*<sup>lox/+</sup> animals. Double heterozygotes were always similar to wild-type mice. Mice were genotyped by PCR.

The following primers were used for genotyping: the conditional *Robo3* allele, 5'-CCA GGG AAA AAC TTG AGG TTG CAG CTA G-3' and 5'-GAT TAG GGG AGG TGA GAC ATA GGG-3', the *Krox20:Cre* allele, 5'-AGT CCA TAT ATG GGC AGC GAC-3' and 5'-ATC AGT GCG TTC GAA CGC TA-3', and the T*au*<sup>GFP</sup> allele, 5'-GAG GGC GAT GCC ACC TAC GGC AAG-3' and 5'-CTC AGG GCG GAC TGG GTG CTC AGG-3'. All PCR had 34 cycles with an annealing temperature of 58℃.

In the *Tau*<sup>GFP</sup> line, upon Cre recombination in neurons, the Stop cassette is excised leading to permanent expression of a myristoylated GFP in axons and of ß-galactosidase in nuclei (*Hippenmeyer et al., 2005*).

## Histology and immunocytochemistry

Mice were perfused transcardially with 4% PFA in 0.12 mM phosphate buffer. Cortices were flattened between microscope slides and post-fixed in 4% PFA and vibratome (Leica Microsystems, Germany) sectioned at 50 µm. Hindbrains and thalamus were post-fixed in 4% PFA, cryoprotected in 30% sucrose and sectioned at 35 µm with a freezing microtome (Microm, France).

For cytochrome oxidase staining (*Melzer et al., 1994*), sections were incubated at room temperature for 24 hours in 10% sucrose, 0.3 g/L cytochrome C from equine heart (Sigma), 0.02 g/L catalase from bovine liver (Sigma-Aldrich, Saint Louis, MO) and 0.25 g/L DAB (Sigma-Aldrich). The endogenous fluorescence of the GFP in *Krox20:Cre;Tau*<sup>GFP</sup> was not affected after treatment and could be imaged on the same sections; however, the GFP signal was further enhanced by immunostaining.

For immunohistochemistry, neonatal and adult brains were processed as described previously (*Marillat et al., 2002*). The following primary antibodies were used: guinea pig anti-Vglut2 (1:1000, Millipore AB-2251), rabbit anti-*β*Gal (1:1000, 55976 Cappel, Cochranville, PA), rabbit anti-c-Fos (1:1000, sc-52 Santa-Cruz biotechnology, Santa Cruz, CA, on sections, or 1:2000, 226–003 Synaptic Systems, Germany, for iDISCO+ studies), rabbit anti-GFP (1:300, A11122 Thermo Fisher, Waltham, MA, on sections), and chicken anti-GFP (1:800, ab13970 Abcam UK on sections or 1:2000, GFP-1020 Aves, Tigard, OR, for iDISCO studies). The following secondary antibodies were used on sections: donkey anti-mouse, anti-rabbit and anti-guinea pig coupled to CY3 or CY5 (1:600, Jackson Laboratories, West Grove, PA), donkey anti-mouse, anti-rabbit and anti-chicken coupled to Alexa Fluor 488, 568 or 657 (1:600, Thermo Fisher) for sections and iDISCO studies. Sections were counterstained with Hoechst 33258 (10 µg/mL, Sigma-Aldrich).

Sections were examined using a fluorescent microscope (DM6000, Leica Microsystems) equipped with a CoolSnapHQ camera (Princeton Instruments, Trenton, NJ), a confocal microscope (FV1000, Olympus, Japan), or a slide scanner (Nanozoomer, Hamamatsu, Japan).

## iDISCO+ processing and light-sheet microscopy

Adult mice or P4, P8 pups were euthanized with a rising gradient of $CO_2$ and fixed with intracardiac perfusion of 4% PFA in PBS. All harvested samples were post-fixed overnight at 4℃ in 4% PFA in PBS.

Fixed samples were washed in PBS for 1 hour twice, then in 20% methanol (in ddH$_2$O) for 1 hour, 40% methanol for 1 hour, 60% methanol for 1 hour, 80% methanol for 1 hour, and 100% methanol for 1 hour twice. Samples were then bleached with 5% $H_2O_2$ (1 vol of 30% $H_2O_2$ for 5 volumes of methanol, ice cold) at 4℃ overnight. After bleaching, samples were re-equilibrated at room temperature slowly and re-hydrated in 80% methanol in $H_2O$ for 1 hour, 60% methanol/$H_2O$ for 1 hour, 40% methanol/$H_2O$ for 1 hour, 20% methanol/$H_2O$ for 1 hour, and finally in PBS/0.2% TritonX-100 for 1 hr twice.

Pre-treated samples were then incubated in PBS/0.2% TritonX-100/20% DMSO/0.3M glycine at 37℃ for 36 hours, then blocked in PBS/0.2% TritonX-100/10% DMSO/6% donkey serum at 37℃ for 2 days. Samples were then incubated in primary antibodies: chicken anti-GFP (1:2000, Aves GFP-1020), rabbit anti-c-Fos (1:2000, Synaptic Systems, 226–003) in PBS-Tween 0.2% with heparin 10 µg/mL (PTwH)/5% DMSO/3% donkey serum at 37℃ for 4 to 7 days. Samples were then washed in PTwH for 24 hours (five changes of the PTwH solution over that time), then incubated in secondary

antibody donkey anti-rabbit-Alexa647 from Thermo Fisher or donkey anti-chicken from Jackson Immunoresearch at 1:500 in PTwH/3% donkey serum) at 37°C for 4 to 7 days. Samples were finally washed in PTwH for 1 day before clearing and imaging. Immunolabeled brains were dehydrated in 20% methanol (in ddH$_2$O) for 1 hour, 40% methanol/H$_2$O for 1 hour, 60% methanol/H$_2$O for 1 hour, 80% methanol/H$_2$O for 1 hour, and 100% methanol for 1 hour twice. Samples were incubated overnight in 1 vol of methanol/2 vol of dichloromethane (DCM, Sigma-Aldrich 270997−12 × 100 ML) until they sank to the bottom of the vial (plastic Eppendorf tubes were used throughout the process). The methanol was then washed for 20 minutes twice in 100% DCM. Finally, samples were incubated (without shaking) in dibenzyl ether (DBE, Sigma-Aldrich 108014–1 KG) until clear (about 30 minutes) and then stored in DBE at room temperature.

Cleared samples were imaged in sagittal orientation (right lateral side up) on a light-sheet microscope (Ultramicroscope II, LaVision Biotec, Germany) equipped with a sCMOS camera (Andor Neo, UK) and a 2X/0.5 objective lens (MVPLAPO 2x) equipped with a 6 mm working distance dipping cap. Version v144 of the Imspector Microscope controller software was used. The microscope is equipped with LED lasers (488 nm and 640 nm) with three fixed light-sheet generating lenses. Scans were made at the 0.8X zoom magnification (1.6X effective magnification), with a light-sheet numerical aperture of 0.1. Emission filters used are 525/50 and 680/30. The samples were scanned with a step-size of 3 µm using the continuous light-sheet scanning method with the included contrast blending algorithm for the 640 nm channel (20 acquisitions per plane with a 50 ms exposure), and without horizontal scanning for the 480 nm channel (50 ms exposure). To speed up the acquisitions, both channels were acquired in two separate scans.

Maximum 3D projections in *Figure 2A* and all panels of *Figures 3*, *4* and *7B* were performed using Imaris (Bitplane, http://www.bitplane.com/imaris/imaris), and generated from manual 3D segmentation of the raw data using the surface tool. ClearMap (*Renier et al., 2016*) (https://www.idisco.info) was used to quantify and register c-Fos+ cells in *Figure 7A*. Parameters were set as previously described, and automated isolation of the cortex was done using scripts available online.

## In situ hybridization

Antisense riboprobes were labeled with digoxigenin-11-D-UTP (Roche Diagnostics) as described previously (*Marillat et al., 2002*) by in vitro transcription of mouse cDNAs encoding *robo3* or an exon specific probe of *robo3* targeting the floxed region (*Renier et al., 2010*).

## DiI tracing

4% PFA fixed P4 pups were injected with small crystals of 1,1'-dioctadecyl-3,3,3',3'-tetramethylindocarbocyanine perchlorate (DiI, Thermo Fisher) and 4-(4-(dihexadecylamino)styryl)-*N*-methylpyridinium iodide (DiA, Invitrogen) using glass micropipettes. For anterograde tracing, the dye crystals were injected unilaterally in the PrV. For retrograde tracing of the PrV nuclei, the cortices were removed to expose the thalamus and DiI or DiA crystals were at the level of the VPM.

Brains were kept at 37°C for 4 weeks. Brains were cut in 80 µm sections with a vibratome (Leica) and counterstained with Hoechst.

## Infraorbital nerve lesions

P0-P1 pups were cold anesthetized, and an incision was made between the whisker pad and the eye. The nerve was cut with scissors under a dissecting scope. The pups were allowed to recover for 10 days and then perfused.

## c-Fos expression and whisker activity

P20-P30 mice were anesthetized with ketamine, and all whiskers were trimmed on the left side. In different experiments, either all whiskers were spared on the right side, or only selected whisker rows or arcs were spared. Mice were allowed to recover from anesthesia for 6 to 12 hours, and then left alone in a large (1 m x 60 cm) 'enriched' cage in the dark for 1 hour before being perfused and processed for c-Fos immunostaining.

## Quantifications and statistical analysis

Areas were calculated with NDPview (Hamamatsu) from cytochrome oxidase staining at P10 or Vglut2 staining when done in adults. For quantification of map areas from P10 flattened cortices, the surface in controls was limited to the first four barrels in row a, four in row b, six in row c, seven in row d, and eight in row e. In *Krox20:Cre;Robo3^{lox/lox}* mutants, the central (ipsi) map area comprises the domain bordered by a thick Vglut2-negative boundary. The peripheral (contra) map area was limited to the barrels located immediately above and below the border of the central ipsi map. To determine individual barrel areas in adults, only the largest unambiguous barrels were measured (first three barrels for rows e, d, c, and first barrels for rows a and b). Areas were assessed on the tangential section showing the most complete map of the PMBSF.

The areas of the VPM nucleus were calculated with NDPview from frontal sections of P4 *Krox20: Cre;Tau^{GFP}* mice stained with cytochrome oxidase and immunostained for GFP, at a mid-level of the VPM, where the barreloid organization was the most obvious.

Results are presented as means ± SEM. Differences of the means between two sample sets were assessed by two-tailed non-parametric Mann-Whitney test. Statistics were carried out with Prism (Graphpad Software, La Jolla, CA).

## Acknowledgements

We thank P Charnay for providing the *Krox20:Cre* line, and S Arber for the Tau^{GFP} mice. We also thank N Narboux-Nême for technical help. FMR laboratory was supported by the Swiss National Science Foundation (31003A_149573) and the Novartis Research Foundation. The teams of AC and PG are part of the Ecole des Neurosciences de Paris training network.

## Additional information

### Funding

| Funder | Grant reference number | Author |
|---|---|---|
| Association Française contre les Myopathies | | Nicolas Renier |
| National Institute of Neurological Disorders and Stroke | RO1 NS039050 | Reha S Erzurumlu |
| Agence Nationale de la Recherche | ANR-08-MNP-030 | Filippo M Rijli Patricia Gaspar Alain Chédotal |
| Swiss National Science Foundation | CRSI33_127440 | Filippo M Rijli |
| Agence Nationale de la Recherche | ANR-08-MNP-032 | Filippo M Rijli Patricia Gaspar Alain Chédotal |
| Agence Nationale de la Recherche | ANR-14-CE13-0004-01 | Filippo M Rijli Patricia Gaspar Alain Chédotal |
| Agence Nationale de la Recherche | ANR-10-LABX-65 | Filippo M Rijli Patricia Gaspar Alain Chédotal |
| Fondation pour la Recherche Médicale | DEQ20120323700 | Alain Chédotal |

The funders had no role in study design, data collection and interpretation, or the decision to submit the work for publication.

### Author contributions

NR, Conceptualization, Data curation, Formal analysis, Investigation, Visualization, Writing—original draft, Writing—review and editing; CD, Data curation, Investigation, Visualization, Writing—review and editing; RSE, Conceptualization, Data curation, Investigation, Methodology, Writing—original

draft, Writing—review and editing; CFK, Data curation, Formal analysis, Investigation, Writing—review and editing; FMR, Data curation, Formal analysis, Supervision, Funding acquisition, Investigation, Methodology, Writing—review and editing; PG, Conceptualization, Data curation, Formal analysis, Supervision, Validation, Investigation, Visualization, Methodology, Writing—original draft; AC, Conceptualization, Data curation, Formal analysis, Supervision, Funding acquisition, Validation, Investigation, Visualization, Methodology, Writing—original draft, Project administration

### Author ORCIDs
Nicolas Renier, http://orcid.org/0000-0003-2642-4402
Claudius F Kratochwil, http://orcid.org/0000-0002-5646-3114
Filippo M Rijli, http://orcid.org/0000-0003-0515-0182
Patricia Gaspar, http://orcid.org/0000-0003-4217-5717
Alain Chédotal, http://orcid.org/0000-0001-7577-3794

### Ethics
Animal experimentation: All animal procedures were carried out in accordance to institutional guidelines and approved by the UPMC University ethic committee (Comité Charles Darwin, authorization # 03787.02). All surgery was performed under ketamine/xylazine anesthesia, and every effort was made to minimize suffering.

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
