## [Decision Letter]

Thank you for submitting your article "A mutant with bilateral whisker to barrel inputs unveils somatosensory mapping rules in the cerebral cortex" for consideration by *eLife*. Your article has been reviewed by two peer reviewers, and the evaluation has been overseen by a Reviewing Editor and Andrew King as the Senior Editor. The following individual involved in review of your submission has agreed to reveal their identity: Denis Jabaudon (Reviewer #2).

The reviewers have discussed the reviews with one another and the Reviewing Editor has drafted this decision to help you prepare a revised submission.

In this manuscript, the authors use an elegant genetic strategy to compromise the trigeminothalamic projection in mice. Sensory information arising from one whisker is mapped onto one barrelette in the brainstem. This study shows that after conditional deletion of Robo3, a receptor mediating midline crossing at the principal sensory nucleus of the trigeminal nerve (PrV), the same information is mapped bilaterally in the secondary relay in the thalamus and in the cerebral cortex. The model is a promising tool to manipulate sensory projections that could help to understand rules governing the formation of sensory maps in general. The data are very robust and well-presented graphically. However, there are certain issues that need to be clarified, especially concerning the focus given to this model in the present manuscript.

Major revisions:

1) Although the experimental paradigm used is novel and interesting, a major concern is that it is difficult to identify a new principle beyond what is already known about the mechanisms underlying somatosensory map formation. For example, the authors attempt to compare their results with existing duplication models of the barrel field, but the contribution of the present study and advances it offers to understanding the rules governing the development of the somatosensory system are unclear. One suggestion is to better articulate on the first/second page of the Introduction what is being tested here – overlap of sensory fields, map-building and/or whether the inputs are segregated if midline crossing is aberrant?

2) One point of interest to the reviewers would be to use the model to better highlight the mechanisms involved in the formation of ipsilateral and contralateral bilateral projections:

A) You found that within the thalamus and cortex the ipsi and contra projections form a distinct and duplicated map. Therefore, you conclude that a mechanism may exist for segregation based on competition between these two subpopulations of axons. The whisker map projects to the thalamus and cortex by both the ipsi and contra axons by proximity rules rather than obeying common gradients at the target. This result is considered "unexpected"; expand on the concept of proximity rules.

B) This result could also be explained by the distinct arrival times of ipsi versus contra axons. In fact, the map organized by the ipsi projections is better defined and arranged within the target, in both the thalamus and cortex. Moreover, rows B and A that are the latest rows to develop are less represented in the contralateral map, suggesting that they are formed later. Therefore, it is possible that ipsi axons arrive to the thalamus first, impose the whisker map representation in place and when contra axons arrive they take the remaining space and impose the second map in an ordered manner. This possibility should be tested or, at least considered and discussed, and if this scenario is true, this implies that competition among these axons is not the primary mechanism.

C) For the ipsi-contra aspect, the proportion of ipsi vs contra axons should be quantified in the *Robo3-cKO*. This would be an extremely important piece of information for understanding the phenotype of the duplication.

3) The role of spontaneous activity and activity-dependent competition is stressed in the process of innervation and mapping. While the study shows that the developmental program controlling the bilateral segregation of visual and auditory inputs is conserved and can be activated in thalamic nuclei for the somatosensory modality (with some differences in continuity of the map), the role of activity is not clear.

A) There are no direct experiments addressing the role of activity-dependence, except for the sensory deprivation paradigm. The last two lines of the Introduction beginning with "the crossed and uncrossed sensory afferents compete for space in both the thalamic relay and the cerebral cortex" are vague. Either add any experiments you have done or tone down this aspect in the manuscript.

B) Rebsam et al., 2009 (J Neurosci. 29(47):14855-63) could be cited as another example of mis-guidance at the midline. In this case, in the absence of the "ipsi" receptor implementing midline avoidance, some retinal ganglion cell axons cross the midline, target to the relatively correct site within the thalamus, but the final pattern of segregation in the normal target site is activity dependent. Compare results, especially in the Discussion, subsection “Orientation and competition of the representations”.

4) Molecular gradients vs. activity based rules:

A) In the subsection "Orientation and polarity of the maps", two possibilities are posed: i). that afferents follow topographic guidance cues determined by molecular guidance cues in the cortex/thalamus, with the outcome of neighboring whiskers of the ipsi and contra map in register and ii). That activity-based rules lead to coordination of afferents such that they cluster together, and that nearest neighbor association is more relevant than molecular gradients. The last sentence of this section does not clearly state which of the two options is supported.

B) It is difficult to discern through the text and figure legends that deafferentation disrupts the segregation of ipsi and contra in the KO.

C) An assumption in the manuscript is that the gradients of molecules for patterning of target areas are unchanged in the *Robo3-cKO*, but has not been tested. Please clarify.

D) You do allude to "tags on the axons" (first paragraph of Discussion and molecular expression profiles in the subsection “Duplication and segregation of whisker maps”). But this comes in the Discussion, and in the latter section, whether these molecular entities would implement the formation of distinct functional units is not considered.

5) One missing point of discussion is whether molecular changes occur when axons cross (or do not cross) the midline, such that they are impaired in their final pattern of connectivity. Michalski et al., on which the senior author of the present paper is listed, and which addresses this issue in a Robo3 KO. The final pattern in the Michalski study, could also be due to molecular affinities of axons with each other (ipsi-contra) or with target cells.

6) Figures: Figure 6 is difficult to comprehend and there seems to be erroneous representation. First, the data in the cKO panels and the quantification seem to be swapped between ipsi and contra, as written in the legend, and to make sense to what has been shown before. Moreover, it lacks the control *Robo3-cKO* intact panels for comparison. This is an essential component of this figure and it should be included.

7) Textual issues:

A) Throughout the manuscript there are sentences that are awkward and difficult to read and understand (e.g., at the end of the long last section: what is the conclusion from this part? And last sentence of the first paragraph of the Discussion).

B) The Discussion in general is complicated, repetitive and without a clear focus. Addressing some of the points in # 1-4 above would help in revising the Discussion.

---

## [Author Response]

*Major revisions:*

*1) Although the experimental paradigm used is novel and interesting, a major concern is that it is difficult to identify a new principle beyond what is already known about the mechanisms underlying somatosensory map formation. For example, the authors attempt to compare their results with existing duplication models of the barrel field, but the contribution of the present study and advances it offers to understanding the rules governing the development of the somatosensory system are unclear. One suggestion is to better articulate on the first/second page of the Introduction what is being tested here – overlap of sensory fields, map-building and/or whether the inputs are segregated if midline crossing is aberrant?*

We thank the reviewers and editor for their appreciation of our results, and for the thoughtful comments and propositions to improve our manuscript. In the present revised version we have extensively modified the Introduction and Discussion to focus on what we believe these mutants illustrate best, namely the rules of sensory map building. In particular, we chose to highlight the insights provided by our study on the importance of nearest-neighbors interactions during map building that prevail over genetic/molecular pre-patterning of the thalamic and cortical anlage. These ideas and the different hypotheses that are at stake are now explicated on the first two pages of the Introduction and the first two pages of the Discussion.

*2) One point of interest to the reviewers would be to use the model to better highlight the mechanisms involved in the formation of ipsilateral and contralateral bilateral projections:*

*A) You found that within the thalamus and cortex the ipsi and contra projections form a distinct and duplicated map. Therefore, you conclude that a mechanism may exist for segregation based on competition between these two subpopulations of axons. The whisker map projects to the thalamus and cortex by both the ipsi and contra axons by proximity rules rather than obeying common gradients at the target. This result is considered "unexpected"; expand on the concept of proximity rules.*

*B) This result could also be explained by the distinct arrival times of ipsi versus contra axons. In fact, the map organized by the ipsi projections is better defined and arranged within the target, in both the thalamus and cortex. Moreover, rows B and A that are the latest rows to develop are less represented in the contralateral map, suggesting that they are formed later. Therefore, it is possible that ipsi axons arrive to the thalamus first, impose the whisker map representation in place and when contra axons arrive they take the remaining space and impose the second map in an ordered manner. This possibility should be tested or, at least considered and discussed, and if this scenario is true, this implies that competition among these axons is not the primary mechanism.*

The reviewers make an important point here, as timing is an important issue that needs to be considered in the interpretation of our results. We thank the reviewing team for raising these questions. In our revised manuscript we expanded the text on these considerations (A and B), and added a paragraph on the timing of arrival of the axons in the Discussion (B). Technically, it is very challenging to compare the exact timing of arrival of ipsilateral versus contralateral axons because there are no constitutive markers for either population, and specifically targeting those projections before E11 with tracers would be quite challenging. However, we now provide a supplementary Figure 2, where such tracing was done at E15. This shows that the ipsi and contralateral trigemino-thalamic axons travel side by side in the lemniscus, and that they are both reach the VPM by E15. Moreover, at E18, the R3-derived trigemino-thalamic axons have all terminated in the VPM, forming what appear to be 2 separate domains in the mutants. Overall, these data support the contention that ipsi and contralateral trigeminal axons reach the VB at the same time in the Robo3-CKO.

*C) For the ipsi-contra aspect, the proportion of ipsi vs contra axons should be quantified in the Robo3-cKO. This would be an extremely important piece of information for understanding the phenotype of the duplication.*

It is very difficult (or impossible) to quantify exactly the proportion of crossed versus uncrossed trigemino-thalamaic axons as for instance GFP is not only expressed in rhombomere 3 by PrV axons. However, we show that there should be about 3.3 times more ipsilateral axons than contralateral ascending from the PrV by quantifying (reported in the text, results section) the proportion of cell bodies retrogradely traced on each side of the PrV from VPM injections, which is an indirect, but fair, measure of the laterality of the projections.

*3) The role of spontaneous activity and activity-dependent competition is stressed in the process of innervation and mapping. While the study shows that the developmental program controlling the bilateral segregation of visual and auditory inputs is conserved and can be activated in thalamic nuclei for the somatosensory modality (with some differences in continuity of the map), the role of activity is not clear.*

*A) There are no direct experiments addressing the role of activity-dependence, except for the sensory deprivation paradigm. The last two lines of the Introduction beginning with "the crossed and uncrossed sensory afferents compete for space in both the thalamic relay and the cerebral cortex" are vague. Either add any experiments you have done or tone down this aspect in the manuscript.*

*B) Rebsam et al., 2009 (J Neurosci. 29(47):14855-63) could be cited as another example of mis-guidance at the midline. In this case, in the absence of the "ipsi" receptor implementing midline avoidance, some retinal ganglion cell axons cross the midline, target to the relatively correct site within the thalamus, but the final pattern of segregation in the normal target site is activity dependent. Compare results, especially in the Discussion, subsection “Orientation and competition of the representations”.*

The reviewers make an important point. Our experiments tested the role of sensory inputs but not neural activity per se. Consequently, we toned down the discussion on the activity dependent mechanisms potentially at play in our study, and added references for the work of Rebsam et al. in the Introduction and Discussion, with more in depth comparisons between the somatosensory and visual systems.

*4) Molecular gradients vs. activity based rules:*

*A) In the subsection "Orientation and polarity of the maps", two possibilities are posed: i). that afferents follow topographic guidance cues determined by molecular guidance cues in the cortex/thalamus, with the outcome of neighboring whiskers of the ipsi and contra map in register and ii). That activity-based rules lead to coordination of afferents such that they cluster together, and that nearest neighbor association is more relevant than molecular gradients. The last sentence of this section does not clearly state which of the two options is supported.*

In the revised version, this has been clarified at the end of the paragraph.

*B) It is difficult to discern through the text and figure legends that deafferentation disrupts the segregation of ipsi and contra in the KO.*

We apologize for this lack of clarity: we do not think that the de-afferentiation affects the segregation of ipsi and contra inputs, or at least we don’t have evidence that it does. We observe fusion of the maps after de-afferentiation and show that the representation of the unlesioned map expands when the competing ipsi or contralateral inputs have been removed by deafferentiation. This has been clarified in the text.

*C) An assumption in the manuscript is that the gradients of molecules for patterning of target areas are unchanged in the Robo3-cKO, but has not been tested. Please clarify.*

This is an important point: we have now added a new panel in Figure 1 showing the expression gradient of ephrin-A5 in the forebrain, which is not affected in the mutant. Discussions concerning this point have been added in the Results section and in the Discussion.

*D) You do allude to "tags on the axons" (first paragraph of Discussion and molecular expression profiles in the subsection “Duplication and segregation of whisker maps”). But this comes in the Discussion, and in the latter section, whether these molecular entities would implement the formation of distinct functional units is not considered.*

This was confusing and has been deleted from the revised manuscript.

*5) One missing point of discussion is whether molecular changes occur when axons cross (or do not cross) the midline, such that they are impaired in their final pattern of connectivity. Michalski et al., on which the senior author of the present paper is listed, and which addresses this issue in a Robo3 KO. The final pattern in the Michalski study, could also be due to molecular affinities of axons with each other (ipsi-contra) or with target cells.*

This is a good point, which has now been added to the Discussion. We also added a reference to the Badura et al. study on the same mutants, analysing the olivo-cerebellar projections. The major difference of our current study with the results from Michalski et al. is that the uncrossing was complete in the case of the auditory system, instead of a mixed crossing phenotype as seen here. Therefore, the Michalski study was uniquely positioned to answer the question of synaptic maturation, whereas our current study would yield a more complex picture, akin to the Badura study, where results on the synaptic maturation at the target could be muddled by the presence of both ipsi and contra inputs in the local network.

*6) Figures: Figure 6 is difficult to comprehend and there seems to be erroneous representation. First, the data in the cKO panels and the quantification seem to be swapped between ipsi and contra, as written in the legend, and to make sense to what has been shown before. Moreover, it lacks the control Robo3-cKO intact panels for comparison. This is an essential component of this figure and it should be included.*

We thank the reviewers for pointing this out: there was indeed a mistake in the ordering of the panels between the ipsi and contra sides of the lesion. This has been corrected. Also, the unlesioned controls have been added to the figure for reference. Thank you for bringing this up and we apologize for this mistake.

*7) Textual issues:*

A) Throughout the manuscript there are sentences that are awkward and difficult to read and understand (e.g., at the end of the long last section: what is the conclusion from this part? And last sentence of the first paragraph of the Discussion).

*B) The Discussion in general is complicated, repetitive and without a clear focus. Addressing some of the points in # 1-4 above would help in revising the Discussion.*

We hope the modifications and rewriting of the manuscript will have helped to solve those issues.